# Tumor Microenvironment Responsive Nanomicelle with Folic Acid Modification Co-Delivery of Doxorubicin/Shikonin for Triple Negative Breast Cancer Treatment

**DOI:** 10.3390/ph16030374

**Published:** 2023-03-01

**Authors:** Wu Zhong, Zhehao Shen, Menglan Wang, Hongyi Wang, Yuting Sun, Xiaojun Tao, Defu Hou

**Affiliations:** 1Key Laboratory of Study and Discovery of Small Targeted Molecules of Hunan Province, School of Medicine, Changsha 410013, China; 2Department of Biochemistry and Molecular Biology, School of Medicine, Hunan Normal University, Changsha 410003, China

**Keywords:** nanomicelles, doxorubicin, shikonin, tumor microenvironment-sensitive, TNBC, targeted therapy

## Abstract

Triple negative breast cancer (TNBC), which has poor prognosis, easily develops drug resistance and metastasizes. In general, those TNBC characteristics are related to a high activation of the epithelial-mesenchymal transition (EMT) pathway, which is inhibited by shikonin (SKN). Therefore, the synergistic therapy of SKN and doxorubicin (DOX) will increase anti-tumor efficacy and reduce metastasis. In this study, we prepared the folic acid-linked PEG nanomicelle (NM) grafted with the DOX (denoted as FPD) to load the SKN. We prepared the SKN@FPD NM according to the effective ratio of dual drugs, where the drug loadings of DOX and SKN were 8.86 ± 0.21% and 9.43 ± 0.13%, with 121.8 ± 1.1 nm of its hydrodynamic dimension and 6.33 ± 0.16 mV of zeta potential, respectively. The nanomaterials significantly slowed down the release of DOX and SKN over 48 h, leading to the release of pH-responsive drugs. Meanwhile, the prepared NM inhibited the activity of MBA-MD-231 cells in vitro. Further in vitro study revealed that the SKN@FPD NM increased the DOX uptake and significantly reduced the metastasis of MBA-MD-231 cells. Overall, these active-targeting NMs improved the tumor-targeting of small molecular drugs and effectively treated TNBC.

## 1. Introduction

Breast cancer remains by far the most common type of tumor in women. Among breast cancer tumors, triple negative breast cancer (TNBC) has poor prognosis, with an 11% 5-year survival rate due to the lack of estrogen receptor (ER), progesterone receptor (PR), and human epidermal growth factor receptor 2 (HER-2) [1]. In general, TNBC is susceptible to drug resistance and tumor formation, which are associated with cancer stem cells [2]. Due to the lack of the above receptors, classic drug treatments have limited therapeutic effects on TNBC. At present, blockers of the epithelial-mesenchymal transition (EMT) pathway against mTOR, such as the everolimus, have been used clinically [3]. Researchers have reported that the cancer stemness is highly related to the activation of the EMT signaling pathway [4,5]. Doxorubicin (DOX), as a first-line small molecule anti-breast cancer drug, easily causes drug resistance by upregulating the EMT pathway. Shikonin (SKN) is a natural bioactive substance existing in the root of the Lithospermum plant and clinically used in the treatment of hepatitis [6,7]. In addition, recent studies have shown that SKN can inhibit tumor growth through multiple pathways, such as the PI3K/AKT/mTOR pathway and the MAPK or FAK pathways [7,8]. Therefore, combined treatment of DOX and SKN can effectively improve the therapeutic effect and prevent drug resistance and metastasis of TNBC. 

Although both drugs have been applied clinically, with minor side effects, DOX possesses dose-related cardiotoxicity. It has been shown that combination therapy can improve its efficacy and reduce the dosage. However, due to the lack of targeting of these drugs, the dosage needed to achieve effectiveness remains high. Nano delivery systems can efficiently solve the above problems through the enhanced permeability and retention (EPR) effect caused by specific sizes (<200 nm) or modifications of specific targets, which allows nanoparticles to effectively target the tumor [9,10,11]. At present, the liposomes are the only types of nano drugs that have been applied in clinical use. Nanomicelles (NMs) exhibit structures and low molecular weights that are similar to those of liposomes, especially polyethylene glycol (PEG) micelles [11,12]. PEG micelles are usually composed of PEG-grafted ligands and drugs, which can provide higher drug loading [13,14] with simple structures. In addition, the stable properties, high repeatability, and easier industrialization of PEG NMs can provide it with high clinical translational value.

In this study, we used folic acid (FA), a biologically derived ligand whose receptor is overexpressed in TNBC as a target. Grafted PEG formed hydration film and reduced the clearance of NMs in vivo (Figure 1). At the same time, the other end of PEG has been linked to DOX to prepare NMs-loaded SKN. In the experiments, we revealed the synthesis, structure, drug loading, and drug release of NMs. Additionally, we have studied the cellular uptake, the tumor cell viability, and metastasis inhibition of NMs in TNBC cells.

## 2. Results

### 2.1. Cytotoxicity and Synergistic Efficacy of Free Doxorubicin (DOX) and Shikonin (SKN)

Firstly, we studied the anti-tumor efficacy of two free drugs in vitro. As shown in Figure 2A–C, CCK-8 Kit was used to detect the cell viability under different treatments, and results showed that the MBA-MD-231 cell viability decreased gradually with the increase in the drug concentration. According to the calculation (Table 1), the IC_50_ of DOX was 0.1356 ± 0.0100 μg/mL, which is smaller than the IC_50_ (0.3434 ± 0.0203 μg/mL) of SKN. Furthermore, the synergistic efficacy of free DOX and SKN was evaluated by the treatments with the two drugs in different combinations, and the synergy scores were calculated using SynergyFinder 2.0. The cell survival rate was reduced to 0.4328 ± 0.068 when the concentrations of DOX and SKN were 0.1 μg/mL and 0.1 μg/mL, while the cell survival rate was 0.2151 ± 0.0267 under the treatment of 0.2 μg/mL DOX and 0.3 μg/mL SKN. As shown in Figure 2D, the DOX and SKN have a certain synergistic effect, with a synergy score of 3.75. Meanwhile, the 1:1 concentration rate of DOX and SKN showed more significant synergies than the others, which provided an effective proportion for our follow-up experiments.

To prepare the small size nanomicelle (NM) with maximum drug loading, we have adjusted the DOX content in the reactants. As shown in Appendix A, the size of those two NMs with high DOX content were not less than 200 nm, while the low content sample was being used in follow-up experiments. According to the reference, the carbodiimide catalyzer will mainly catalyze γ-carboxylic of FA, and only a few α-carboxylic groups react with other groups [15,16,17]. We have demonstrated the synthesis of nanomaterials by FTIR and ^1^H NMR. As depicted in Figure 3, Folic acid-grafted polyethylene glycol carboxyl (FA-PEG-COOH) exhibited an ether bond C–O–C stretching vibration peak at 1105 cm^−1^, and a polyamide and free carbonyl stretching vibration peak at 1685 cm^−1^ and 1718 cm^−1^ from FA (Appendix A). At the same time, on the ^1^H NMR spectrum, the 8.65 ppm signal of FA and FA-PEG-COOH resulted from heterocycle (Appendix A), whereas the 7.65 ppm signal and the 6.64 ppm signal of FA and FA-PEG-COOH resulted from the hydrogen on benzene ring, and the 6.94 ppm signal came from the secondary amine of the ortho substitution on the benzene ring. These results demonstrated the successful synthesis of FA-PEG-COOH. For folic acid-grafted polyethylene glycol-doxorubicin (FA-PEG-DOX), the FITR spectrum proves the 1281 cm^−1^ vC-O peak of the phenol, and the strong 1728 cm^−1^ carbonyl stretching vibration peak in the DOX demonstrated the DOX grafting. Meanwhile, 5.82 ppm, 5.34 ppm, 5.05 ppm, and 4.12 ppm were the unreacted hydroxyl signals or hydroxyl o-carbon signals, and the 4.00 ppm signal was the characteristic methyl peak from DOX.

### 2.2. Preparation and Characterization of NMs

Referring to the synergy of two free drugs in Section 2.1, we selected 1:1 concentration rate of the DOX and SKN to prepare the NMs and verified them by DLS and TEM. As depicted in Figure 4, the hydrodynamic dimension of FA-PEG-DOX nanomicelle (FPD NM) was 106.8 ± 0.8 nm, with its PDI at 0.226 ± 0.009, while the hydrodynamic dimension of SKN-loaded FA-PEG-DOX nanomicelle (SKN@FPD NM) was 121.8 ± 1.1 nm, with its PDI at 0.216 ± 0.020. Compared with the blank NM, the drug-loaded NM increased in size, which could also be confirmed according to the TEM. Additionally, the TEM images suggested regular shape and uniform size. Meanwhile, the two NMs have similar zeta potential. The zeta potential of the FPD NM was −6.44 ± 0.23 mV, while that of the SKN@FPD NM was −6.33 ± 0.16 mV. Additionally, FPD NM exhibits high stability within 3 days at low temperatures (Appendix A). The data above indicate the successful preparation of NMs. 

### 2.3. Drug Loading and Release of NMs

To detect the drug loading and concentration of NMs, an ultraviolet visible light absorption instrument was used to detect concentrations of DOX and SKN. The content of DOX in the FA-PEG-DOX was 9.78 ± 0.14%, according to the formula in Section 4.7. Meanwhile, the measured drug loadings of DOX and SKN in SKN@FPD NM were 8.86 ± 0.21% and 9.43 ± 0.13%, respectively. The total drug loadings of SKN@FPD NM were near 20%, which is relatively high in micellar nanoparticles. We subsequently performed drug release experiments. As depicted in Figure 5, free DOX and SKN released quickly in PBS (pH = 7.4) within the first 8 h, with rates of 67.57 ± 1.45% and 72.12 ± 2.61%, respectively. However, the free DOX showed a significant increase in the release of acidic PBS (pH = 5.5), with the rate of 77.36 ± 1.85%, while the rate of free SKN was at 65.32 ± 2.08% of the free SKN. For the NMs, DOX was bound to the PEG, and showed a slow release with the rate of 38.56 ± 1.05%. At the same time, the release of FA-PEG-DOX increased slightly in the acidic PBS, with the rate being 43.96 ± 1.42. The acid environment significantly increased the release rates of SKN from 50.11 ± 1.33% to 71.72 ± 1.65%, indicating that the NMs can effectively delay drug release under physiological conditions, but an increased drug release in the tumor micro environment or tumor endosome.

### 2.4. Cytotoxicity of NMs In Vitro

The toxicities of the nano drug and the free drug to TNBC in vitro were detected using a CCK-8 Kit. As depicted in Figure 6, the cell survival rate was 0.5706 ± 0.0283 at the maximum DOX concentration, while it was 0.6394 ± 0.0202 at the maximum FA-PEG-DOX concentration. The PEG-bond seems to have no significant influence on the toxicity of DOX. Additionally, the survival rate for dual free drugs was 0.4252 ± 0.0246 at the largest concentration. Compared with the survival rate for dual free drugs, the survival rate for the NMs was smaller, at the rate of 0.3730 ± 0.0096. These results indicate that the SKN@FPD NM we prepared exhibit higher anti TNBC activities under the same drug concentration than do the free drugs.

### 2.5. NMs Increase the Cellular Uptake In Vitro

To illustrate the relationship between cellular uptake and toxicity, we have continued to capture the fluorescent images of tumor cells after treatment with free drugs and NMs. As depicted in Figure 7, free DOX showed a continuous uptake during the whole 4 h. Free dual drugs seemed to show a similar DOX uptake to that of the free DOX, and most of the free DOX enters into the nucleus. Compared with the free drugs, SKN@FPD NM had a higher continuous cellular uptake during the 4 h. Additionally, the DOX in FPD was partly distributed in the cytoplasm near the nucleus, which formed a dim red light around the nucleus. The above data show that the grafted FA-PEG can significantly increase the cell uptake of DOX, maintaining a toxic concentration of DOX in the cytoplasm of the tumor.

### 2.6. NMs Inhibit Cell Migration In Vitro

To depict the inhibition of the synergistic drugs and SKN@FPD NM, wound healing assays were performed. As shown in Figure 8, the control group cells exhibited obvious metastasis at 6 h and 12 h. In contrast, those cells treated by free DOX had no notable metastasis during the 12 h period. At the same time, the dual drug group and the NMs group showed almost no metastasis compared with the DOX group or control group during the 12 h period. Additionally, at 6 or 12 h after treatment with synergistic drugs and SKN@FPD NM, the cell morphology of the MDA-MB-231 cells had changed significantly. In conclusion, we provided an active-targeting NM for the TNBC treatments by the inhibition of cell metastasis.

## 3. Discussion

Currently, monotherapy has been replaced by synergistic therapy due to the low risk of drug resistance, metastasis, and the recurrence of synergistic therapy in triple negative breast cancer (TNBC) treatment. Different from other breast cancers, TNBC cells lack other receptors, such as the estrogen receptor (ER), progesterone receptor (PR), and human epidermal growth factor receptor (HER-2), which have been proven to be important targets in other breast cancer treatments [18]. Therapies based on some newly discovered blockers, or the aforementioned therapies that are also synergistic with other traditional small molecule drugs, have been studied in clinical research, significantly changing TNBC treatment [19,20]. Immune checkpoint blockers, such as PD–1/PD-L1 blockers, show unusually brilliant results for TNBC treatment [21]. In addition, AstraZeneca and Daiichi Sankyo jointly announced the TROP2 blocker Datopotamab Deruxtecan, an antibody-conjugated EMT blocker with a high disease control rate of 95%, showing the potential prospect for the application of EMT blockers in TNBC treatment [22]. Most of the EMT pathways are relatively upstream in the cells, and treatments without effective tumor targeting will lead to a high occurrence of side effects [23]. Thus, it is necessary to improve drug targeting to reduce undesirable side effects.

The nano delivery system is currently one of the delivery systems that can effectively solve the above problems. Selecting a specific size (100–200 nm) for the nanoparticles, as well as target grafting, will lead to a strong anti-tumor targeting [24]. For polyethylene glycol (PEG) nanomicelles (NMs), both ends of PEG generally bond to drugs and ligands for effective targeting [25]. In this study, we synthesized a pH-sensitive PEG NMs, grafted with folic acid (FA) and doxorubicin (DOX), that realized controlled release and increased cell uptake of the drugs. Generally, amide bonds and ester bonds are stable in aqueous solution for several days at low temperatures. In all the experiments, we did not find any phenomena indicating that the nanomaterials were unstable. The 4-Dimethylaminopyridine (DMAP) and 1-Ethyl-3-(3-dimethylaminopropyl) carbodiimide hydrochloride (EDCI) in the reaction system are easily soluble in water as ureide, a major by-product of EDCI catalytic reaction that was significantly reduced when DMAP was added. This means that impurities can be removed. In this research, we have prepared three kinds of NMs with different DOX ratios. Two types of folic acid grafted polyethylene glycol-doxorubicin (FA-PEG-DOX) with high DOX ratios have much larger hydrodynamic dimension than that of the lowest DOX FA-PEG-DOX. Therefore, the FA-PEG-DOX ratio of 1:10 had been chosen for follow-up experiments. Then, we tested the Fourier-transform infrared spectroscopy (FTIR) and nuclear magnetic resonance hydrogen spectrum (^1^H NMR), and the results showed the successful synthesis of nanomaterials.

To choose an appropriate DOX:SKN ratio to prepare maximum efficacy NMs, we first selected the synergistic experiment of free drugs. The results showed that the 1:1 ratio of DOX:SKN exhibited the greatest anti-tumor efficacy. Therefore, this ratio of DOX:FA-PEG-DOX was used for the NMs. Both the hydrodynamic dimension of prepared FA-PEG-DOX nanomicelle (FPD NM) and SKN loaded FA-PEG-DOX nanomicelle (SKN@FPD NM) were less than 200 nm, and the SKN@FPD NM (121.8 ± 1.1 nm) was larger than the FPD NM (106.8 ± 0.8 nm) due to the SKN loaded. At the same time, the free carboxyl from the folic acid led to a negatively charged surface of the NM. The SKN@FPD NM had a similar zeta potential (−6.33 ± 0.16 mV) compared with FPD NM (−6.44 ± 0.23 mV) because of the SKN exhibited neutral pH value in an aqueous solution. Additionally, we observed that the release of SKN increased significantly at pH 5.5. Compared with 50.11 ± 1.33% of pH 7, the release of SKN in NM increased to 71.72 ± 1.65 at pH 5.5 over a 48 h period. DOX contains numerous carbonyl and hydroxyl groups and possesses an amino group, while a free carboxyl was noted in FA. FA carboxyl is more easily ionized and becomes more hydrophilic when in a PBS environment (pH = 7.4). When the pH dropped to 5.5, the DOX groups were more likely to accept protons and increase hydrophilicity, while the FA carboxyl ionization was weakened, and the hydrophobicity was enhanced. This leads to the instability of NMs and the quick release of drugs.

Next, we tested the anti-tumor efficacy in vitro. The results showed that the grafted FA-PEG did not significantly reduce the efficacy of DOX. SKN@FPD NM treatment exhibited higher efficacy than that of the free dual drugs. Further wound healing experiments proved that the synergistic treatment of dual drugs almost completely inhibited TNBC metastasis. Free DOX also inhibits tumor metastasis to a certain extent, but continuous treatment may increase the tumor stem cells, leading to metastasis or recurrence. There is no difference in the inhibition of tumor metastasis between free dual drugs and SKN@FPD NM. To further study the uptake of nanoparticles in TNBC cells, we prepared the cellular uptake experiment. The results showed that SKN@FPD NM could be absorbed into the cytoplasm when it efficiently entered into the nucleus. The cellular uptake of DOX and both free drugs increased slightly with time during the experimental period. In contrast, the intracellular SKN@FPD increased significantly, indicating that the NMs we prepared achieved effective TNBC cell targeting.

## 4. Materials and Methods

### 4.1. Materials

Doxorubicin (DOX), Folic acid (FA), Dimethyl sulfoxide (DMSO), 1,1’-Carbonyldiimidazole (CDI) and 4-Dimethylaminopyridine (DMAP) were obtained from Aladdin (Shanghai, China); Amino polyethylene glycol carboxyl (NH_2_-PEG-COOH, 2000 Da) and 1-Ethyl-3-(3-dimethylaminopropyl) carbodiimide hydrochloride (EDCI) were obtained from Macklin Biochemical Co., Ltd. (Shanghai, China); Cell Counting Kit-8 (CCK-8) and 4’,6-Diamidino-2-phenylindole (DMAP) were obtained from Beyotime Biotechnology Co., Ltd. (Nantong, China); DMEM-H, and fetal bovine serum were obtained from Thermofisher (Shanghai, China).

### 4.2. Cell Culture

Triple negative breast cancer cells (MDA-MB-231) were seeded into a 25T culture flask in DMEM containing 10% fetal bovine serum and 1% penicillin-streptomycin solution. The cells were cultured in a carbon dioxide incubator (37 °C and 5% CO_2_), and the trypsin-containing EDTA was used for passage.

### 4.3. Anti-Tumor Efficacy and Synergy of Drugs In Vitro

MDA-MB-231 cells were seeded into 96-well plates (5000 cells per well). DOX and SKN with different concentrations were scattered in the complete culture medium and the cells were treated for 2 days. Then, the complete fresh culture medium with 10% CCK-8 reagent replaced the above medium, and the absorbance at 490 nm was then detected after 1 h of treatment. Similar methods were used to evaluate the efficacy of free drugs and nanoparticles. The cell viability was calculated as follows:$$\mathrm{Inhibition}\;\mathrm{rate}\;\left(\%\right)=\frac{\mathrm{OD}(\mathrm{Experimental}\;\mathrm{Group})-\mathrm{OD}(\mathrm{Blank}\;\mathrm{group})}{\mathrm{OD}(\mathrm{Control}\;\mathrm{group})-\mathrm{OD}(\mathrm{Blank}\;\mathrm{group})}\;\times \;100\%$$
where the 10% CCK-8 medium added to the well with no cells was called the blank group, while the same CCK-8 medium added to the well without drug treatment was called the control group.

To explore the synergistic efficacy of DOX and SKN in TNBC therapy, SynergyFinder3.0 was used to calculate the synergy score of the two drugs, according to methods used in our previous research.

### 4.4. Synthesis of Nanomaterials

The synthesis of folic acid grafted polyethylene glycol carboxyl (FA-PEG-COOH) was performed according to methods in a previous study [12]. In short, 1 g of FA and 1 g of EDCI were added into 30 mL DMSO and activated at 40 °C for 30 min. Then 2 g of NH_2_-PEG-COOH was added, and reacting for 2 d. The above solution was then dialyzed (3000 Da) in deionized water for 6 h and further freeze-dried.

Folic acid-grafted polyethylene glycol-doxorubicin (FA-PEG-DOX) was synthetized similar to the methods in reference [12]. A total of 0.1 g of DOX with 0.20 g of CDI, 0.2 g of DOX with 0.4 g of CDI, and 0.3 g of DOX with 0.6 g of CDI were dissolved in 10 mL DMSO and stirred at 40 °C for 4 h, respectively. 1.0 g of FA-PEG-COOH was dissolved in the DMSO with 0.1 g of DMAP and 0.2 g of EDCI and activated for 30 min until mixing. Then, the mixed solution was stirred and reacted at 40 °C for 1 day. Finally, the products were obtained by dialyzing (3000 Da) in deionized water for 6 h and freeze-drying the mixed reaction solution.

### 4.5. Preparation and Characterization of Nanomicelle

A total of 10 mg of FA-PEG-DOX was dissolved in 5 mL of DMSO and dialyzed (7000 Da) for 4 h to obtain the FA-PEG-DOX nanomicelle (FPD NM). The concentration ratio with strong synergistic efficacy (the mass ratio was 1:1 of DOX:SKN) was used to prepare SKN@FA-PEG-DOX nanomicelle (SKN@FPD NM), whose method was similar to the preparation of FPD NM. The hydrodynamic dimension and the zeta potential of both NMs were measured using a dynamic light scattering instrument (hydrodynamic dimension: wavelength of 658 nm, temperature of 25 ± 0.1 °C, and DLS angle of 90°; zeta potential: 1.4 v/cm, 13.0 mA, and 25 ± 0.1 °C). For transmission electron microscope (TEM) scanning, both NMs were dropped on the copper mesh for air-drying. Then, they were photographed on a TEM.

### 4.6. Detection of Nanomaterials

Fourier transform infrared spectroscopy (FTIR) and nuclear magnetic resonance spectroscopy (^1^H NMR) were used to study the structure of the nanomaterials. Samples were ground and pressed into pieces with KBr to test the FTIR, t, and dissolved in deuterated DMSO to test the ^1^H NMR. The infrared spectra of samples were obtained by scanning in the range of 400~4000 cm^−1^.

### 4.7. Drug Loading and Drug Release In Vitro

To test the drug loading, we first tested the DOX degree of substitution in FA-PEG-DOX. For standard curve drawing, hydrochloric acid (HCl) solutions (pH = 5.5) with the DOX concentrations of 2, 4, 6, 8, and 10 μg/mL were prepared, and their absorptions were detected on an ultraviolet visible light absorption instrument at 480 nm, respectively. The absorption of 20 μg/mL of FA-PEG-DOX HCL solution (pH = 5.5) was detected at 488 nm, and the substitution of DOX was calculated as follows:$${w\%}_{\mathrm{DOX}}=\frac{C_{\mathrm{DOX}}}{C_{\text{FA-PEG-DOX}}}\times \;100\%$$

Dual-wavelength spectrophotometry was used to detect the dual drug concentration in SKN@ FPD NM. The standard curves were drawn by detecting the 480 nm and 565 nm absorptions of dual drugs, and the drug loadings were calculated as follows:$${\mathrm{LC}}_{\mathrm{SKN}}\%=\frac{C_{\mathrm{SKN}}}{C_{\mathrm{SKN}}+C_{\text{FA-PEG-DOX}}}\times \;100\%$$
$${\mathrm{LC}}_{\mathrm{DOX}}\%=\frac{C_{\text{FA-PEG-DOX}}}{C_{\mathrm{SKN}}+C_{\text{FA-PEG-DOX}}}\times \;{w\%}_{\mathrm{DOX}}\;\times \;100\%$$

To test the drug release of SKN@FA-PEG-DOX NM (SKN@FPD NM), 5 mL of NM was placed into the dialysis bag (7000 Da) and dialyzed in 30 mL PBS (pH = 7.4 or 5.5) on a 37 °C shaker (75 rpm). The obsolete PBS was replaced by 30 mL of fresh PBS at 0, 0.5, 2, 4, 8, 16, 24, and 48 h, and then the pH of the obsolete PBS was adjusted to 5.5. The absorptions of the obsolete PBS were detected at 480 nm and 565 nm, and the release of the dual drugs was calculated as follows:$$Q_{t}\%=\frac{V_{0}C_{0}+\sum {\;}_{t=0}^{n}V_{t}C_{t}}{V_{\mathrm{db}}C_{\mathrm{db}(\mathrm{Drug})}}\times \;100\%$$
where **Q_t_** is the drug release rate at t hour; **V_t_** and **C_t_** are the volumes of PBS and drug concentration in PBS at t hour out of the dialysis bag, respectively; **V_db_** is the volume of PBS in the dialysis bag, and **C_db(Drug)_** is the initial concentration of the nanocomposite sample (t = 0, 0.5, 1, ⋯, *n*, ⋯, 48 h; both V_0_ and C_0_ are equal to 0).

### 4.8. Cell Migration Assay

MDA-MB-231 cells were seeded into 24-well plates (5 × 10^5^ cells per well). DOX, DOX + SKN and SKN@ FPD NM with different concentrations were added to culture those cells once they grew to 80%. The cells were photographed at 0 h, 6 h, and 12 h (100×), the mobility of which was calculated using the Image J software Version 1.53t.

### 4.9. In Vitro Intracellular Uptake

MDA-MB-231 cells were seeded into 24-well plates (2 × 10^5^ cells per well). DOX + SKN and SKN@FPD NM with different concentrations were added to culture those cells once they grew to 50%. The treated cells were dyed with DAPI (1 μg/mL) and photographed at 2 h, 4 h, and 6 h, and then the cellular uptake was calculated using the Image J software.

### 4.10. Statistical Analysis

All groups contained 3-fold parallel data and were expressed as MEAN ± SD; differences between groups were t-tested by GRAPHPAD 7.0. The data were drawn by ORIGIN 2017 and GRAPHPAD 7.0. For the differences between two groups, * *p* < 0.05 and ** *p* < 0.01 were considered significant, and *** *p* < 0.001 and **** *p* < 0.0001 were considered highly significant.

## 5. Conclusions

Novel nanomicelle (NMs) were synthesized from FA, PEG, and DOX to increase the targeting and stability of DOX and SKN. Amide reaction was used to synthesis the nanomicelle materials, while the FTIR and ^1^H NMR proved the synthesis of the nanomaterials. Appropriate size blank NM (~100 nm) were prepared by adjusting the ratio of DOX to FA-PEG-COOH, while the best drugs ratio (1:1) to prepare SKN loaded NMs (SKN@FPD NM) was obtained through the free drug synergistic experiment. Compared with FPD NM, the size of SKN@FPD NM increased. However, they exhibited similar shapes and zeta potentials. The dual drug loading of obtained SKN@FPD NM was near 20%, which is higher in the nanomicelles. At the same time, the prepared NMs could effectively reduce the drug release in PBS with pH 7.4. However, due to the exchange of hydrophobic and hydrophilic ends at different pH levels, the NM achieved acid responsiveness, and drug release increased significantly under acidic conditions. The cell experiments in vitro showed that the NM realized long-lasting cellular uptake compared with that of the free drugs. Moreover, the NM significantly inhibited TNBC viability, which was better than that of the DOX combined with SKN. The free dual drugs and NM exhibited a robust inhibition of tumor migration, which was better than that of free DOX. In short, SKN@FPD NM is a promising nanomicelle for the treatment of TNBC.

## Figures and Tables

**Figure 1 pharmaceuticals-16-00374-f001:**
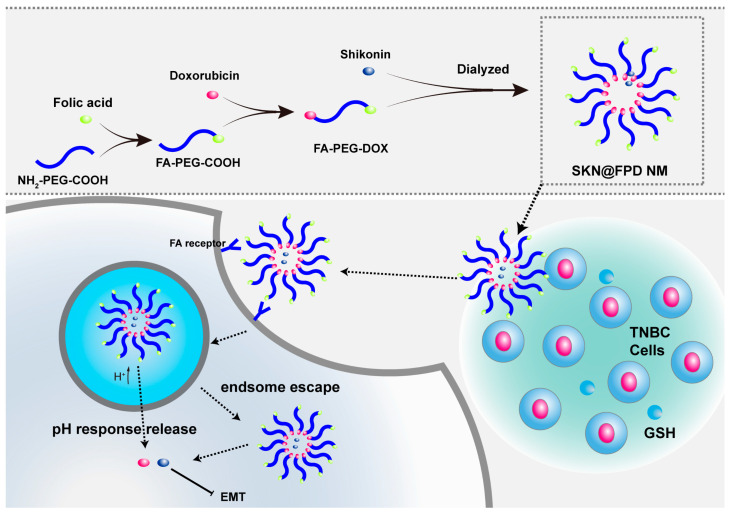
Synthetic route of the SKN@FPD NMs and the mechanism scheme of the SKN@FPD NM in vitro.

**Figure 2 pharmaceuticals-16-00374-f002:**
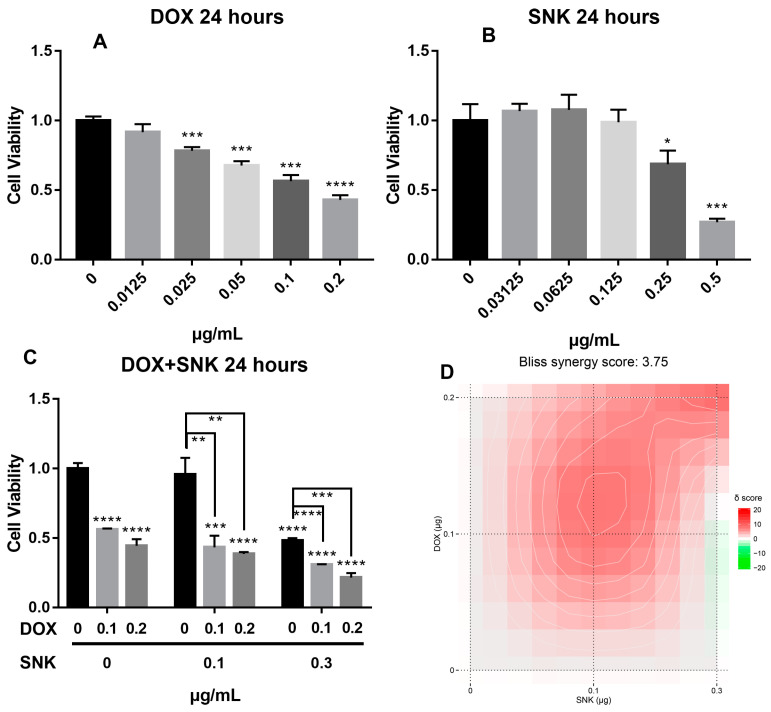
Toxicity of DOX (**A**), SKN (**B**), and dual drugs (**C**) in MDA-MB-231. (**D**) Combination synergy score analysis using SynergyFinder 2.0. * *p* < 0.05, ** *p* < 0.01, *** *p* < 0.001, and **** *p* < 0.0001 represent significant differences.

**Figure 3 pharmaceuticals-16-00374-f003:**
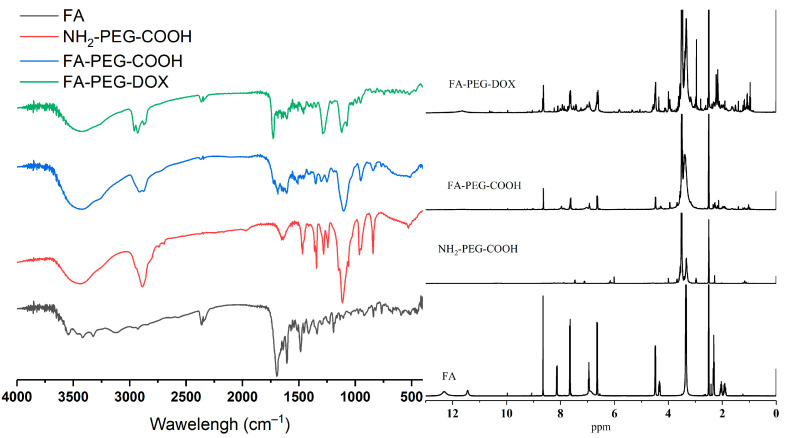
Synthesis of nanomaterials. FTIR of FA, NH_2_-PEG-COOH, FA-PEG-COOH, and FA-PEG-DOX (**left**) and ^1^H NMR of FA, NH_2_-PEG-COOH, FA-PEG-COOH, and FA-PEG-DOX (**right**).

**Figure 4 pharmaceuticals-16-00374-f004:**
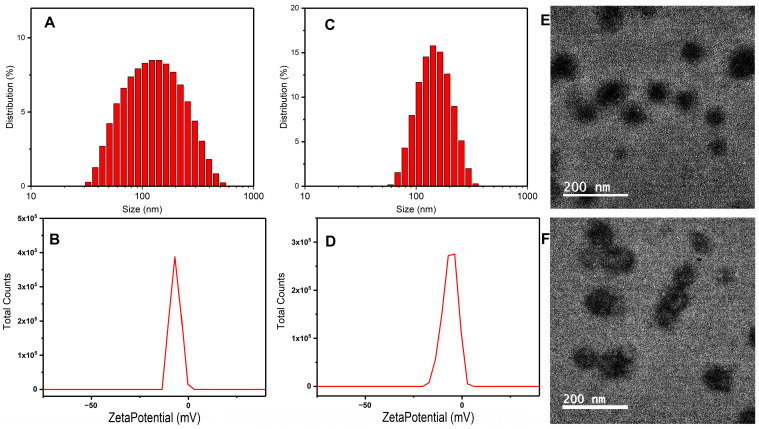
Characterization of FPD and SKN@ FPD NMs. Size of FPD NPs (**A**), SKN@ FPD NMs (**C**), and zeta potential FPD (**B**), and SKN@ FPD NMs (**D**) measured by DLS. TEM of FPD NMs (**E**) and SKN@ FPD NMs (**F**).

**Figure 5 pharmaceuticals-16-00374-f005:**
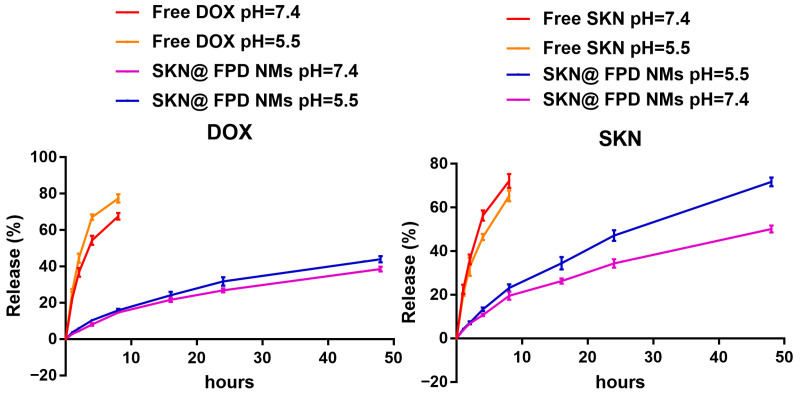
DOX (left) and SKN (right) release of SKN@PPD NPs in PBS (pH = 7.4) and acid PBS (pH = 5.5) over 48 h.

**Figure 6 pharmaceuticals-16-00374-f006:**
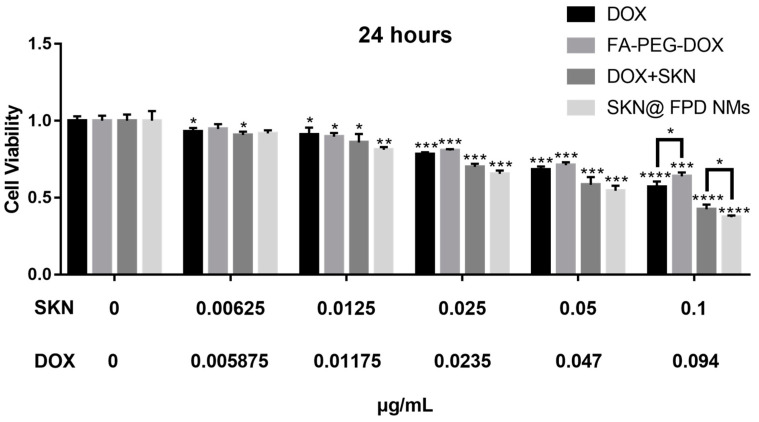
Cell toxicity of DOX, FA-PEG-DOX, free dual drugs, and SKN@FPD NMs in MDA-MB-231 cells. * *p* < 0.05, ** *p* < 0.01, *** *p* < 0.001, and **** *p* < 0.0001 represent significant differences.

**Figure 7 pharmaceuticals-16-00374-f007:**
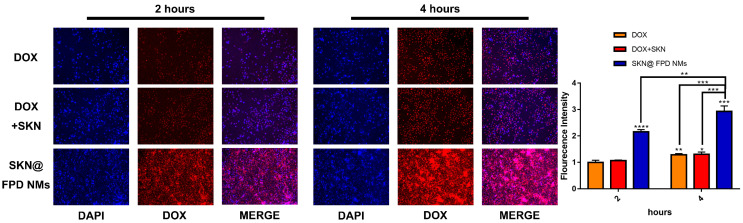
Cell uptake of free DOX, free dual drugs, and SKN@FPD NMs in MDA-MB-231 cells, with the DOX concentration of 3 μg/mL and the DAPI concentration of 1 μg/mL. * *p* < 0.05, ** *p* < 0.01, *** *p* < 0.001, and **** *p* < 0.0001 represent significant differences.

**Figure 8 pharmaceuticals-16-00374-f008:**
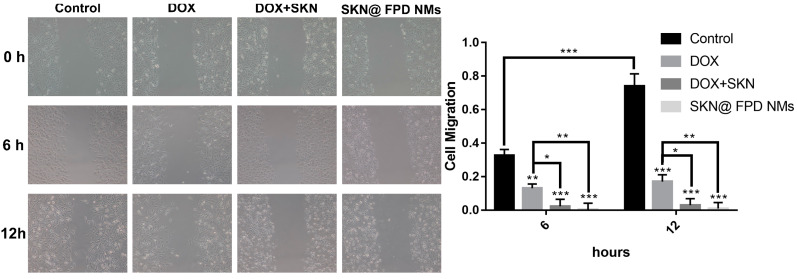
Wound healing experiments using the MDA-MB-231 cells after treatment with DOX, free dual drugs, and SKN@FPD NMs. * *p* < 0.05, ** *p* < 0.01, *** *p* < 0.001.

**Table 1 pharmaceuticals-16-00374-t001:** IC50 of DOX and SKN.

Drugs	IC50 (μg/mL)
DOX	0.1356 ± 0.0100
SKN	0.3434 ± 0.0203

## Data Availability

The authors declare that all data supporting the findings of this study are available within the paper.

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
