# Peer review of "Tumor Microenvironment Responsive Nanomicelle with Folic Acid Modification Co-Delivery of Doxorubicin/Shikonin for Triple Negative Breast Cancer Treatment"

_pharmaceuticals, 2023, doi:10.3390/ph16030374_

Round 1
Reviewer 1 Report
The paper deals with the preparation and characterization of responsive nanomicelle functionalized with folic acid for the co-delivery of doxorubicin/shikonin for triple negative breast cancer treatment.
The approach presented in the article is interesting and quite innovative.
Some parts are difficult to follow and should be improved. Below you can find my considerations:
1. Figure2: the concentrations of B are different from those of A. Is it possible to make a comparison?
2. Figure 5: In the text the authors refer to CKN, while in the figure they refer to SKN. Throughout the article there is some confusion regarding the abbreviations.
3. The article does not mention the aggregation phenomenon typical of nanosystems. Have tests been performed to determine whether nanomicelles tend to aggregate over time under different experimental conditions?
4. Insert a table with the IR and NMR data, this could ameliorate the understanding in the discussion of the spectra
5. Line 210: NMR instead MNR
6. Line 211: “to prepare” instead “for prepared”
7. Line 213-216: This part is absolutely not clear! The authors should make a scheme in which the compounds and their respective acronyms are reported. For example onn line 215, the acronym FPD NMs appears without describing what they are.
8. Line 214: “preparation” instead “prepared”
9. Line 219: CKN?
10. Line 222: “pH” instead “Ph”
11. Line 226: in vitro in italics
12. Line 229: please insert a reference in the sentence “Synthesis of FA-PEG-COOH was referring to the previous study”
13. Line 281: What is PBA-PEG-DOX? It has never been mentioned before.
14. Line 282: Which are the appropriate concentrations? Pleas describe better this experimental part.
15. Line 284: What is the correct abbreviation? SKN@FA-PEG-DOX nanomicelles (SKN@FPD NM)? Abbreviations in the article must be uniform and unambiguous
16. Line 299: What buffer did the authors use to obtain a pH of 5.5?
17. Line 301-302: English is not correct
18. Line 311: Is the cut off of the dialysis membrane used 7000kDa or 3000kDa? Moreover, in the text it appears 3000 da, I don't think it's correct.
Overall the article is interesting, but the authors need to make an effort to improve the presentation. In many parts it is confusing making it very difficult to read. Inserting figures and tables could improve understanding of the data.
Author Response
The paper deals with the preparation and characterization of responsive nanomicelle functionalized with folic acid for the co-delivery of doxorubicin/shikonin for triple negative breast cancer treatment.
The approach presented in the article is interesting and quite innovative.
Some parts are difficult to follow and should be improved. Below you can find my considerations:
- Figure2: the concentrations of B are different from those of A. Is it possible to make a comparison?
Thank you for taking time reading our manuscript. In Figure 2, we have used DOX and SKN with the maximum concentration of 0.2 μg/mL and 0.5 μg/mL, and it is not possible to make a comparison between the two maximum concentrations drugs. Therefore, we have separately calculated the IC50 of DOX and SKN in TNBC cells that can make a comparison. In the latest manuscript, we have added a table about IC50 to explain it.
- Figure 5: In the text the authors refer to CKN, while in the figure they refer to SKN. Throughout the article there is some confusion regarding the abbreviations.
We feel sorry for these mistakes. Thank you for your suggestions. In the latest manuscript, we have revised the mistakes.
- The article does not mention the aggregation phenomenon typical of nanosystems. Have tests been performed to determine whether nanomicelles tend to aggregate over time under different experimental conditions?
Thanks for your suggestion. We have tested the hydrodynamic dimension after 48 hours and 72 hours at 4℃, and added them to the Figure S2 of supplementary data.
- Insert a table with the IR and NMR data, this could ameliorate the understanding in the discussion of the spectra
Thanks for your suggestion. We added two tables of characteristic peaks assignment of FTIR and 1H NMR to the supplementary data.
- Line 210: NMR instead MNR
We feel sorry for these mistakes. Thank you for your suggestions. In the latest manuscript, we have revised the mistakes.
- Line 211: “to prepare” instead “for prepared”
Thank you for your suggestions. In the latest manuscript, we have revised it.
- Line 213-216: This part is absolutely not clear! The authors should make a scheme in which the compounds and their respective acronyms are reported. For example on line 215, the acronym FPD NMs appears without describing what they are.
Thanks for your suggestion, we have added a list of abbreviation after the Conclusion. And we have also added the full meaning before the abbreviation when it was first time appearing at a section.
- Line 214: “preparation” instead “prepared”
Thank you for your suggestion. In the latest manuscript, we have revised it.
- Line 219: CKN?
Thank you for your suggestion. In the latest manuscript, we have revised it.
- Line 222: “pH” instead “Ph”
Thank you for your suggestion. In the latest manuscript, we have revised it.
- Line 226: in vitro in italics
Thank you for your suggestion. In the latest manuscript, we have revised it.
- Line 229: please insert a reference in the sentence “Synthesis of FA-PEG-COOH was referring to the previous study”
Thank you for your suggestion. We have added a reference in this sentence.
- Line 281: What is PBA-PEG-DOX? It has never been mentioned before.
We feel sorry for these mistakes. In the latest manuscript, we have revised the text error.
- Line 282: Which are the appropriate concentrations? Pleas describe better this experimental part.
Thank you for your suggestion. We have added the feeding ratio in this part.
- Line 284: What is the correct abbreviation? SKN@FA-PEG-DOX nanomicelles (SKN@FPD NM)? Abbreviations in the article must be uniform and unambiguous
Thanks for your suggestion. SKN@FPD NM is the abbreviation of SKN@FA-PEG-DOX nanomicelle. And we have revised all the uncorrect abbreviation in the manuscript.
- Line 299: What buffer did the authors use to obtain a pH of 5.5?
Thanks for your suggestions. The HCl was used to obtain a pH of 5.5, and we have added it in the latest manuscript.
- Line 301-302: English is not correct
Thanks for your suggestion. We have revised it in the latest manuscript.
- Line 311: Is the cut off of the dialysis membrane used 7000kDa or 3000kDa? Moreover, in the text it appears 3000 da, I don't think it's correct.
We feel sorry for these mistakes. Thanks for your advice. The 3000 da and 7000 da are correct, we have revised them in the manuscript. Moreover, we used the 7000 da dialysis membrane there.
Reviewer 2 Report
The authors describe the formulation of a novel NMs-delivery of a target-active drug for the treatment of triple-negative breast cancer.
The authors discuss a new approach to the treatment of TNBC ..... Overall, these actively targeted NMs provide a new strategy for the treatment of TNBC. ..... I disagree about "originality," as both the PEG "nano-device" and the drugs used have already been combined with DOX for the same purpose and associated with the active target.
That said, I believe that the association with SKN and the choice of folic acid as the active target are well-studied. The authors study by appropriate methods and comprehensively describe the synthesis, structure, drug loading, and release of NMs. In addition, cellular uptake, tumor cell viability, and metastasis inhibition by NMs in TNBC cells are adequately demonstrated methodologically. Moreover, the results are comprehensive, although non-fluent English, does not always make them easy to interpret.
1)Biodegradable PEG-PCL Nanoparticles for Co-delivery of MUC1 Inhibitor and Doxorubicin for the Confinement of Triple-Negative Breast Cancer Akanksha Behl, et al J Polym Environ. 2022 Nov 11: 1–20. DOI: 10.1007/s10924-022-02654-4
2) Biomimetic peptide display from a polymeric nanoparticle surface for targeting and antitumor activity to human triple-negative breast cancer cells Eric M. Bressler,et al; .J Biomed Mater Res A. 2018 Jun; 106(6): 1753–1764
3) Advanced and Innovative Nano-Systems for Anticancer Targeted Drug Delivery. Lu Tang,1,2,† Jing Li,1,2,† Qingqing Zhao,1,2 Ting Pan,1,2 Hui Zhong,3,* and Wei Wang1,2,* Pharmaceutics. 2021 Aug; 13(8): 1151. doi: 10.3390/pharmaceutics13081151
Author Response
Overall the article is interesting, but the authors need to make an effort to improve the presentation. In many parts it is confusing making it very difficult to read. Inserting figures and tables could improve understanding of the data.
Thank you for taking time giving your suggestions. We have added more table to improve understanding in the latest manuscript.
The authors discuss a new approach to the treatment of TNBC ..... Overall, these actively targeted NMs provide a new strategy for the treatment of TNBC. ..... I disagree about "originality," as both the PEG "nano-device" and the drugs used have already been combined with DOX for the same purpose and associated with the active target.
That said, I believe that the association with SKN and the choice of folic acid as the active target are well-studied. The authors study by appropriate methods and comprehensively describe the synthesis, structure, drug loading, and release of NMs. In addition, cellular uptake, tumor cell viability, and metastasis inhibition by NMs in TNBC cells are adequately demonstrated methodologically. Moreover, the results are comprehensive, although non-fluent English, does not always make them easy to interpret.
1)Biodegradable PEG-PCL Nanoparticles for Co-delivery of MUC1 Inhibitor and Doxorubicin for the Confinement of Triple-Negative Breast Cancer Akanksha Behl, et al J Polym Environ. 2022 Nov 11: 1–20. DOI: 10.1007/s10924-022-02654-4
2) Biomimetic peptide display from a polymeric nanoparticle surface for targeting and antitumor activity to human triple-negative breast cancer cells Eric M. Bressler,et al; .J Biomed Mater Res A. 2018 Jun; 106(6): 1753–1764
3) Advanced and Innovative Nano-Systems for Anticancer Targeted Drug Delivery. Lu Tang,1,2,† Jing Li,1,2,† Qingqing Zhao,1,2 Ting Pan,1,2 Hui Zhong,3,* and Wei Wang1,2,* Pharmaceutics. 2021 Aug; 13(8): 1151. doi: 10.3390/pharmaceutics13081151
Thanks for your suggestions. We have revised this part to make it reflected our actual contribution. Moreover, we have improved the grammar to make it easy to understand.